# *Punica granatum* L. (Pomegranate) Extracts and Their Effects on Healthy and Diseased Skin

**DOI:** 10.3390/pharmaceutics16040458

**Published:** 2024-03-26

**Authors:** Jovana Dimitrijevic, Marina Tomovic, Jovana Bradic, Anica Petrovic, Vladimir Jakovljevic, Marijana Andjic, Jelena Živković, Suzana Đorđević Milošević, Igor Simanic, Nina Dragicevic

**Affiliations:** 1Department of Pharmacy, Faculty of Medical Sciences, University of Kragujevac, 69 Svetozara Markovica St., 34000 Kragujevac, Serbia; dijovana@gmail.com (J.D.); jovanabradickg@gmail.com (J.B.); petkovicanica0@gmail.com (A.P.); 2Center of Excellence for Redox Balance Research in Cardiovascular and Metabolic Disorders, 69 Svetozara Markovica St., 34000 Kragujevac, Serbia; drvladakgbg@yahoo.com; 3Department of Physiology, Faculty of Medical Sciences, University of Kragujevac, 69 Svetozara Markovica St., 34000 Kragujevac, Serbia; 4Department of Human Pathology, Sechenov First Moscow State Medical University, 8 Trubetskaya St., 119991 Moscow, Russia; 5Institute for Medicinal Plants Research “Dr. Josif Pancic”, Tadeusa Koscuska 1, 11000 Belgrade, Serbia; jelenazivkovic1@yahoo.com; 6Environment and Sustainable Development, Singidunum University, Danijelova 32, 11000 Belgrade, Serbia; sdjordjevicmilosevic@singidunum.ac.rs; 7Specialized Hospital for Rehabilitation and Orthopedic Prosthetics, Sokobanjska 17, 11000 Beograd, Serbia; dr.igorsimanic@yahoo.com; 8Department of Physical Medicine and Rehabilitation, Faculty of Medical Sciences, University of Kragujevac, 69 Svetozara Markovica St., 34000 Kragujevac, Serbia; 9Department of Pharmacy, Singidunum University, Danijelova 32, 11000 Belgrade, Serbia; ndragicevic@singidunum.ac.rs

**Keywords:** *Punica granatum*, active compounds, dermatology, cosmetics, skin diseases, skincare

## Abstract

The aim of this review is to provide a summary of the botany, phytochemistry and dermatological effects of *Punica granatum* (PG), with special emphasis on therapeutic mechanisms in various skin conditions. PG peel contains the highest levels of chemical compounds. Due to the high abundance of polyphenolic compounds, including phenolic acids, anthocyanins and flavonoids, exhibiting strong antioxidant properties, PG peel possesses significant health-promoting effects. Up until now, different parts of PG in the form of various extracts, fixed seed oil or individual active compounds have been investigated for various effects on skin conditions in in vitro and in vivo studies, such as antioxidant, anti-inflammatory, antimicrobial, chemoprotective and antiaging effects, as well as positive effects on striae distensae, skin repair mechanisms, erythema, pigmentation and psoriasis. Therefore, formulations containing PG active compounds have been used for skincare of diseased and healthy skin. Only a few effects have been confirmed on human subjects. Based on encouraging results obtained in in vitro and animal studies about the numerous substantial dermatological effects of PG active compounds, future perspectives should incorporate more in vivo investigations in human volunteers. This approach can aid in identifying the optimal concentrations and formulations that would be most efficacious in addressing specific skin conditions.

## 1. Introduction

Throughout history, the use of medicinal products prepared from various herbal species for treating dermatological conditions has been widely documented. Herbal remedies have a long history of use for treating skin problems in the human population and animals from ancient times, which has been well documented in Europe from the Renaissance era [1]. Dermatologic conditions rank as the fourth most prevalent contributor to human diseases. In the Global Burden of Disease (GBD) Study, it has been recorded that approximately one-third of the world population is affected by at least one skin disease and that the estimated burden of these conditions is very large in both high- and low-income countries [2]. Natural products and their active compounds have garnered significant interest in the fields of dermatology and cosmetology, attributed to their proven beneficial effects on the skin, as demonstrated in prior research endeavors [3].

Although Punica granatum (PG) has been extensively studied for its numerous positive effects on overall health, there are limited data and no comprehensive reviews regarding its effects specifically on skin health and the underlying mechanisms. Therefore, this review seeks to consolidate existing knowledge about PG’s impact on skin health and highlight its potential applications in various skin conditions, despite the lack of intensive research in this area (Figure 1).

### 1.1. Taxonomy

In the taxonomic classification of botanical entities, *Punica granatum* belongs to the kingdom *Plantae*, characterized by its designation as plants. Further classification places it in the subkingdom *Tracheobionta*, indicating its categorization among vascular plants. Within the superdivision *Spermatophyta*, the entity is specifically identified as a seed plant. Moving further in the hierarchy, it falls under the division *Magnoliophyta*, a category encompassing flowering plants. Its class is denoted as *Magnoliopsida*, indicating its classification as a dicotyledon. The more specific subclass is *Rosidae*, followed by the order Myrtales. In terms of familial classification, the entity belongs to *Punicaceae*, known as the *Pomegranate* family. The genus is specified as *Punica* L., signifying its botanical grouping as pomegranate. Finally, at the species level, it is identified as *Punica granatum* L. [4].

### 1.2. Botany

*Punica granatum* L. (commonly referred to as pomegranate, from the Latin “pome” which means apple, and “granate” meaning many-seeded) is a small shrub tree that is native to the Mediterranean regions. It belongs to the family *Lythraceae* (previously named *Punicaceae*). PG is characterized by a grow length of 4–5 m, thorny branches around the plant and flaky bark with shiny crumpled petal leaves [5,6].

This small deciduous tree is known for its red-brown bark that becomes grey over time [7]. The specific twisted wooden brown-colored bark can grow up to 5 m tall [6]. There are often a few suckers growing around the PG bark, which need to be removed frequently. The trunk of PG, varying from drooping to erect positions, is round-shaped with alternate open branches that are often spiny, stiff and angular and sometimes prickly at the apex [7].

The leaves of PG are 2–9 cm long and 3 cm wide, with lateral buds on their axils, reddish when young and bright green when matured. The terminal bud can occasionally become thorny and can transform into a flower or cluster of flowers, or it can fall off [7]. Leaves possess a short stem, and their arrangement is described as five to six leaves in a cluster with five to six leaves on branches [8]. PG has red to red-orange funnel-shaped flowers with ornamental collections that can have double or various flowers that are not fully grown for fruit production. Flowers are positioned on short branches more than one year old and can occur alone or in clusters that consist of two to five flowers [7]. PG flowers are 3 cm wide and possess many stamens and pointed sepals. These flowers are described to be located at the peripheral region of each PG branch, which contains two to seven flowers [8].

Fruits of PG grow on short spurs and have a round shape with a crowned base [7]. This base represents a thick tubular calyx. The width of the PG fruit varies from 6 to 12 cm and the average weight is around 200 g. The fruit of this plant is edible, characterized by leathery rind and containing around 600 arils. Arils represent the casings for the PG seeds [8]. The leathery peel is yellow-colored and covered with a light or deep pink layer. The inside of PG fruit has a membranous wall with bitter tissue components that separate sacs, which are filled with sweet acid red- or pink-colored pulp and arils [7]. Wild PG plants possess smaller fruit and arils, as well thicker rinds and higher acidity, in comparison with the cultivated ones [9].

Because of the adaptive nature of PG, it can grow in various regions worldwide with different types of climate conditions, such as the Mediterranean regions of Asia, Africa, America and Europe [7] (Figure 2 and Figure 3).

### 1.3. Active Compounds

PG contains various types of phytochemical compounds. The methanol extract of the bark portion disclosed the presence of polyphenols (such as flavonoids, tannins and phenolic acids), sterols and polyterpenes. Polyphenols (mainly flavonoids) were the most abundant chemical compounds in this plant. Previous investigations proved the presence of polyphenols in the methanolic extract of PG bark in the concentration of 272.82 ± 3.05 µg/mL and flavonoids in the concentration of 38.26 ± 1.78 µg/mL [10]. A high concentration of flavonoid compounds was also shown for other parts of the PG plant [11].

The isolation of active compounds from PG peel is enabled through the use of various types of solvents, including water, ethanol, acetone, chloroform and petroleum ether. The concentration of active compounds has been found to be the highest in ethanol and aqueous extracts [12]. Phytochemicals such as carbohydrates, tannins, alkaloids, flavonoids, saponin, quinones, phenols, terpenoids, cardiac glycosides, coumarins and steroids were isolated from aqueous and ethanol extracts, with the highest concentration detected in the ethanol extract [12]. In a previous investigation, there were different types of phytochemicals detected in the PG peel extract, including ellagitannins, with the most dominant compound being punicalin, followed by punicalagin and phenolic acids (gallic acid and ellagic acid) [13]. In another study, it has been shown that the PG peel has the highest concentrations of gallic and ellagic acids, while punicalagins a and b contributed most to the phenolic content in pomegranate juice [14]. Ellagic acid, gallic acid and punicalagins a and b, obtained from the ethyl acetate fraction of the ethanol extract from the entire fruit, demonstrated potential inhibition of lipopolysaccharide (LPS)-induced nitric oxide (NO), prostaglandin E2 (PGE-2) and interleukin-6 (IL-6) production [15]. Thus, these compounds may play a role in the anti-inflammatory properties associated with PG. The active compound content of the PG peel extract and PG juice are shown in Table 1.

Due to the high concentration of polyphenolic compounds, phenolic acids, anthocyanins, and flavonoids as potent antioxidants, the ethanol extract of PG peel can possess antioxidant, anti-diabetic, anti-obesity and anti-hypertensive traits [12]. Furthermore, the ethanol extract of PG peel has shown antimicrobial activity against two Gram-negative and ten Gram-positive bacteria [16]. Other systemic activities that have been proven in investigations of the effects of PG peel are antimalarial, antitoxic and antigenotoxic effects, as well as the aforementioned anti-inflammatory effect [6].

## 2. Dermatological Effects of *Punica granatum*

From ancient times, PG has been used for treating skin inflammation in the Middle East, India and Iran. Ayurvedic medicine uses different parts of PG to nourish and to restore the balance of the skin [17]. The dermatological effects of various fractions of PG are demonstrated in different in vivo and in vitro models. PG shows different effects, and due to these different activities, it can be successfully used for skincare of diseased skin (e.g., in eczema due to its anti-inflammatory effect, psoriasis, striae, bacterial and fungal infections, etc.), as well as of healthy skin (preventing UV-induced photoaging and skin cancer, preventing chrono-aging, improving wrinkle appearance, etc.).

### 2.1. Antioxidant Activity

Fixed oil derived from PG seeds, known for its high content of free fatty acids, phenolic compounds and phytosterols, particularly linoleic acid (29%) and oleic acid (10%) [18], was investigated for its antioxidant and anti-inflammatory properties in a study conducted in 1999 [19]. Analysis of PG juice fermented with *Saccharomycs bayanus* and cold-pressed PG seed oil revealed significant antioxidant activity, surpassing that of red wine. PG seed oil exhibited a notable 37% inhibition of cyclooxygenase (COX), while fermented PG juice did not inhibit COX. Furthermore, PG seed oil showed superior effects against lipoxygenase (LOX) compared to fermented PG juice, highlighting its anti-inflammatory potential [19]. These findings underscore the significance of PG fractions’ unique composition and therapeutic potential.

Positive effects of the PG fruit extract standardized to 30% punicalagins and 2.3% ellagic acid were noticed against hydrogen peroxide-induced oxidative stress and cytotoxicity in human keratinocyte HaCaT cells. In this in vitro study, researchers measured reactive oxygen species (ROS) and levels of caspases in cell samples, viability of hydrogen peroxide-treated cells and detected apoptosis or necrosis of cells by flow cytometry. Researchers observed that PG fruit extract at concentrations of 12.5, 25 and 50 μg/mL notably decreased hydrogen peroxide-induced ROS production by 1.36-, 1.07- and 1.03-fold, respectively, in the examined HaCaT cell cultures. PG fruit extract and its active compounds punicalagin and ellagic acid possess promising antioxidant effects by decreasing the level of hydrogen peroxide-induced apoptosis and downregulation of the levels of caspase-3 and caspase-7 in these cells [20].

In a recent in vitro study [21], the antioxidant potential of PG seed and peel extracts was evaluated using various solvents for the extraction. Extraction of PG peel using methanol yielded the highest concentration of phenolic compounds and exhibited superior antioxidant activity, with a recorded value of 82%. This extract also demonstrated significant free radical scavenging (81%) and lipid peroxidation prevention (56%) abilities. Additionally, hydroxyl radical scavenging potential was assessed, with the methanol extract of PG peel showing the highest activity (58%). Furthermore, the methanol extract of PG peel exhibited dose-dependent antioxidant effects against Low-Density Lipoprotein (LDL) oxidation, suggesting its promising therapeutic potential in combating oxidative stress. These findings underscore the importance of solvent selection in isolating antioxidant-rich compounds from PG peel, which emerges as a potent source of antioxidants within the plant [21]. The antioxidant effects of PG are summarized in Table 2, including the use of antioxidant properties of PG in cosmetics [22]. It is also worth mentioning that antioxidants in skin products have positive effects both on skin health and on formulation stability [23].

### 2.2. Anti-Inflammatory Activity

An in vitro study that analyzed the anti-inflammatory activity of active compounds of PG fruit isolated with the use of ethyl acetate evaluated the effects of ellagic acid, gallic acid, punicalagin A and punicalagin B in the macrophage cell line RAW264.7 against various inflammatory agents. The reported results revealed cytotoxic effects on lipopolysaccharide (LPS)-induced inflamed RAW264.7 cells, a dose-dependent inhibition of nitrite production (with the highest inhibitory effect measured with the use of ellagic acid), and a dose-dependent suppression of PGE-2 and IL-6 production. However, no suppression of COX-2 gene expression in macrophages was detected. The results of this study indicate the anti-inflammatory potential of the analyzed PG fraction due to high concentrations of ellagic acid, gallic acid, punicalagin A and punicalagin B that possess inhibitory effects on the release of certain proinflammatory mediators (Table 3) [15].

### 2.3. Antimicrobial Activity

An in vivo study [24] investigated the antifungal efficacy of PG peel extract at concentrations of 125 µg/mL, 250 µg/mL and 500 µg/mL, compared to nystatin, in *Wistar* rats with induced oral candidiasis. Treatment involved daily application of the samples for 15 days. In vitro microdilution assays revealed complete inhibition of *Candida albicans* growth with the 250 µg/mL PG peel extract and over 50% inhibition with the 125 µg/mL concentration, demonstrating its efficacy without adverse effects. Notably, significant reductions in *C. albicans* growth were observed after 5, 10 and 15 days of treatment with the PG peel extract, with the 500 µg/mL concentration showing the highest effectiveness and achieving a 100% cure rate after 15 days. Light microscopy examination revealed normal tongue morphology post-treatment, confirming the PG peel extract’s efficacy in oral candidiasis management, comparable to nystatin [24].

In a study [25], PG peel and seed water extracts were employed to enhance the antibacterial effectiveness of chitosan–gold nanoparticles against antibiotic-resistant bacteria. The combination exhibited lower minimum inhibitory concentration (MIC) and minimum bactericidal concentration (MBC), indicating enhanced antimicrobial efficacy and synergistic effects against resistant strains. Time-dependent growth inhibition assays revealed delayed Methicillin-resistant *Staphylococcus aureus* (MRSA) growth with higher concentrations of chitosan–PG extract–gold nanoparticles, with significant inhibition observed at 15.6 μg/mL. The authors proposed that the *Punica granatum* extract’s antimicrobial activity might be linked to its diverse constituents, such as flavonoids, phenolic acids, tannins and others [25].

In a recent study, the antimicrobial effects of the PG peel extract, derived by the application of the supercritical fluid extraction, have been investigated. Using the diffusion disc method, the authors reported the inhibition of the growth of Gram-negative bacteria (*Pseudomonas fluorescens*, *Pseudomonas aeruginosa*, *Escherichia coli)*, Gram-positive bacteria (*Bacillus cereus*, *Staphylococcus aureus* and *Streptococcus pyogenes*) and several fungal species such as *Aspergillus flavus*, *C. albicans*, *Penicillium cyclopium* and *Trichoderma viride*. Furthermore, with the use of the broth microdilution method, the microbial growth inhibition rate for different extract concentrations was determined. The authors reported a significant inhibition of the growth rate of *Pseudomonas fluorescens*, *Pseudomonas aeruginosa*, *Escherichia coli*, *Bacillus cereus* and *Staphylococcus aureus* with the use of 2.7 mg/mL PG peel extract. This study demonstrated the importance of the extraction method based on the existence of active ingredients in the extract, as well as the antimicrobial potential of the PG peel extract on various analyzed pathogens [26]. The antimicrobial effects of PG are summarized in Table 4, including its efficacy when used in cosmetics [27].

Furthermore, certain components of PG, in the form of new herbal antimicrobial products, can be used as skin-related or dental materials [28].

## 3. The Use of *Punica granatum* in Different Skin Conditions

### 3.1. Effects on Dandruff

Another study investigated the effectiveness of the combined use of 28.1% (*w*/*w*) PG flower methanol extract and other plant extracts (including *Rosmarinus officinalis* L., *Matricaria chamomilla* L., *Urtica dioica* L., *Mentha piperita* L. and *Salvia officinalis* L. methanol extracts) in the treatment of chronic dandruff, which is characterized as the inflammation of scalp epidermis caused by Malassezia fungus (Table 4). This study included 30 human subjects. Researchers reported significant dandruff reduction (43%) in 15 subjects during the first week of using the antidandruff shampoo containing aforementioned extracts, and complete dandruff removal after two weeks in the same group of subjects. Dandruff removal in the other 12 subjects was accomplished after 4 weeks and total dandruff removal in the remaining 3 patients was achieved after 5 weeks of treatment. These results indicate the positive effects of the flower extract on dandruff removal, as an efficient and significantly safer option in comparison to the chemical compounds used for this indication [27].

### 3.2. Effects on Acne Vulgaris

Acne vulgaris, a dermatological ailment, results from the interaction of numerous factors. These include hyperkeratosis of sebaceous follicles, the stimulating influence of androgens on sebaceous glands and the inflammatory effects of *Propionibacterium acnes*. The convergence of these elements leads to the development of acne lesions not only on facial skin but also on other areas of the body [29].

Effects of the PG extract on acne vulgaris were investigated in several models that simulate pathogenic mechanisms of this condition. A 70% PG acetone extract and four isolated active compounds, hydrolysable tannins, of PG were tested. Punicalagin, punicalin, strictinin and granatin were isolated with the use of column chromatography, which was combined with in vitro anti-inflammatory-guided fractionation. In an animal model inducing ear edema in *Wistar* rats, the authors observed substantially reduced edema in the rat ears, decreased inflammatory status and a lower rate of *P. acnes* growth, in the group of rats treated with the ointment containing the PG extract. Unlike retinoic acid and salicylic acid, which can cause skin irritation and desquamation, none of these side effects were reported in rat skin after performing a single-dose skin irritation test with the PG extract. Further in vitro analysis confirmed strong antibacterial activity of the PG extract against *P. acnes*, with the diameter of the inhibition zone ranging from 11.3 to 17.1 mm, and also against *S. aureus* with the diameter of the inhibition zone ranging from 12.6 to 15.9 mm. The antimicrobial activity of four hydrolysable tannins was further investigated by using a bioguided fractionation–isolation system, indicating that punicalagin and punicalin had significant antibacterial activities against *P. acnes* and *S. aureus*, while strictinin A and granatin B were less effective. Researchers reported several mechanisms of antibacterial effects of PG polyphenols, including shrinkage and damage of bacterial surface and biofilm formation of *P. acnes* and *S. aureus* and inhibition of lipase activity. Another in vitro analysis was conducted in HaCaT cells with testosterone-induced epithelial cells and keratin accumulation, where four hydrolysable tannins at 50 µg/mL significantly decreased cell proliferation, indicating the ability of these compounds to decrease keratinocyte over-proliferation, one of the crucial processes in acne formation. Thus, the PG extract can potentially be used as a growth regulator of skin keratinocytes. Moreover, the authors reported protective effects of the PG extract against UV-induced oxidative stress in the HaCaT cell model and the reduction of photoaging markers, which indicates its potential use for skin protection from UV radiation. The anti-inflammatory effects of the PG extract and its compounds were also shown in an LPS-treated RAW 264.7 cell model. Heat-killed *P. acnes* (HKP) was used instead of LPS for the simulation of irritation and inflammation caused by *P. acnes*. Each of the four hydrolysable tannins exhibited nitric oxide inhibitory effects exceeding 50%, and notably suppressed approximately 50% of prostaglandin E2 production in the HKP-induced RAW 264.7 cell model. Dose-dependent inhibition of IL-8 and TNF-α in a human monocytic cell line was also reported, and granatine B showed the strongest inhibitory activity, which indicates its anti-inflammatory cytokine effects (Table 5) [30].

### 3.3. Chemoprotective Effects and Effects against UV Radiation-Induced Skin Damage

Skin cancer ranks among the most prevalent human cancers. The majority of non-melanoma skin cancers can be readily treated through surgical removal, but there are high-risk melanomas associated with significant morbidity and mortality. One of these high-risk cases is melanoma with histologic and clinical characteristics displaying aggressiveness, which has a high potential for local recurrence and regional or distant metastasis. High-risk cases require adjuvant treatment and many strategies using different active compounds are being developed [31].

Chemoprotective effects of cold-pressed PG seed oil were investigated in vivo in 7,12-dimethylbenzanthracene (DMBA)- and 12-O-tetradecanoylphorbol 13-acetate (TPA)-induced skin tumor in mice. A topical preparation that contained 5% PG seed oil in 100 µL acetone solution was used in the study. After initiating carcinogenesis and 20 weeks of treatment, the results revealed that topical treatment with the PG seed oil did not inhibit the onset of tumor formation. However, it had a significant effect on tumor development rate. Namely, the incidence of the skin tumor 20 weeks after the initiated carcinogenesis was 100% in the control group and 93% in the group of mice treated with the PG seed oil. Thus, this study confirmed the chemoprotective effects of the PG seed oil by the inhibition of prostaglandin synthesis, which contributes to the inhibition of ornithine decarboxylase and the suppression of skin cancer promotion (Table 6) [32].

The effects of the PG fruit extract (acetone/water = 70:30 *v*/*v*) on the modulation of mitogen-activated protein kinases, NF-κB pathways and inhibition of skin tumorigenesis were analyzed using an animal model (CD-1 mice). The topical treatment of 2 mg PG fruit extract in 200 µL acetone/mouse and 2 mg PG fruit extract in 100 µL acetone/mouse had a significant protective effect in mice against several TPA-mediated processes, including cutaneous edema, epidermal hyperplasia, epidermal ornithine decarboxylase activity, COX-2 protein expression and phosphorylation of mitogen-activated protein kinases. These results suggested that the PG fruit extract possesses an anti-tumor-promoting activity, due to the suppression of both traditional and newly identified biomarkers associated with induced tumor promotion (Table 6) [33].

In the field of photodermatology, UV light is considered to be the component of the solar system with the highest risk of skin damage. Therefore, numerous strategies have been developed to achieve effective skin protection against UVA and UVB radiation [34].

UVA radiation extends from 320 to 400 nm and can cause oxidative damage that leads to inflammation, genetic mutation and immunosuppression, all of which are critical factors in tumor advancement (Table 6) [35].

**Table 6 pharmaceutics-16-00458-t006:** Chemoprotective effects and effects against UV radiation-induced skin damage of *Punica granatum*.

*Punica granatum* Formulation	Effect	Model	Material	Dosage/Concentration	Mechanism	Study
Seed oil	Chemoprotective effects	DMBA- and TPA-induced skin tumor	CD1 mice	5%	Inhibition of prostaglandin synthesis, which contributes to inhibition of ornithine decarboxylase and skin cancer promotion	Hora et al. [32]
Fruit extract	Chemoprotective effects	TPA-induced tumorigenesis	CD1 mice	2 mg	Modulation of mitogen-activated protein kinases and NF-κB pathways	Afaq et al. [33]
Fruit extract	Photochemopreventive activity	UVA-induced tumorigenesis	NHEK cells	60, 80, 100 pg/mL	inhibition of UVA-mediated phosphorylation, ERKU2, phosphorylation of AKT1 at Ser473 and phosphorylation of STAT3, mTOR and p70S6K; inhibition of UVA-mediated increase in PCNA and Ki-67 protein expression; augmentation of UVA-mediated cell cycle arrest; dose-dependent modulations in levels of antiapoptotic protein Bcl-2 family	Syed et al. [35]
Ellagic acid	Antioxidative and photoprotective effects	UVA-induced oxidative stress	HaCaT cells	25–75 µM	Inhibition of UVA-induced generation of ROS	Hseu et al. [36]
Juice extract, oil	Photoprotective effects	UVB-mediated skin cell damage	3D full-thickness human reconstituted skin (EpiDerm™ FT-200 (Mattek Corp. (Ashland, MA, USA))	extract (5–10 μg), oil (1–2 μL)	Inhibition of CPD and 8-OHdG formation, UVB-mediated protein carbonyl group increase and cell proliferation; increase in tropoelastin levels; inhibition of UVB-mediated increase in the protein levels and activity of MMPs and phosphorylation of c-jun and expression of c-Fos	Afaq et al. [37]
Fruit extract	Photochemoprotective effects	UVB-mediated skin damage	SehcanismsKH1 mice	0.2%, w/vol	Antioxidant, anti-inflammatory, antiproliferative and DNA repair mechanisms	Khan et al. [38]
Fruit extract	Prevention of UVB skin damage and photoaging	UVB-mediated oxidative stress	HaCaT cells	10–40 μg/mL	Inhibition of UVB-mediated cytotoxicity, decrease in glutathione and TIMP1 levels; increase in LPO and MMP protein expression; inhibition of UVB-mediated phosphorylation of c-jun and MAPK	Zaid et al. [39]
Peel extract	Photoprotective effects	In vitro evaluation of SPF	Mansur method	1 mg/mL	Absorption of UVA and UVB photons, antioxidant and antiaging activity	Zeghad et al. [40]
Peel extract	Protective effects against UV radiation- induced carcinogenesis	UV-induced skin cancer	SKH1/CRL mice	41.9 mg/kg	Inhibition of mutated p53 and PCNA expression	Gómez-García et al. [41]
Fruit extract incorporated in solid lipid nanoparticles	Anticancer effects	Cancer cell culture model	HFB-4, MCF-7, PC-3 cells	49.2 μg/mL–219 μg/mL	Anti-invasive, anti- proliferative, proapoptotic, and anti-inflammatory activity	Badawi et al. [42]
Fruit extract incorporated in solid lipid nanoparticles in transdermal emulgel	Anticancer effects	Induced Ehrlich ascites carcinoma	*Swiss albino* female mice	1.2 mg/mL	Hydrolysable tannins and ellagic acid constraining tumor cell growth	Teaima et al. [43]

The effects of the PG fruit extract (acetone/water = 70:30 *v*/*v*) on the UVA radiation-induced activation of cellular pathways in normal human epidermal keratinocytes (NHEKs) were analyzed. The authors used the PG fruit extract in different concentrations, i.e., 60, 80 and 100 pg/mL, dissolved in dimethyl sulfoxide. One of the reported effects of PG fruit extract on UVA-exposed NHEK cell culture was dose-dependent inhibition of the UVA-mediated phosphorylation of extracellular signal-regulated kinase (ERKU2), serine-threonine protein kinase *AKT1* at Ser^473^ and Signal transducers and activators of transcription 3 (STAT3). An inhibition of UVA-mediated phosphorylation of mammalian target of rapamycin (mTOR) and its downstream target p70S6K was also reported. Moreover, researchers observed a reduction in UVA-induced upregulation of proliferating cell nuclear antigen (PCNA) and Ki-67 protein expression. Furthermore, an enhancement of UVA-induced cell cycle arrest and dose-dependent alterations in levels of the antiapoptotic protein Bcl-2 family, accompanied by a simultaneous increase in Bax and Bad protein expression, were noted. The authors reported potent antioxidant and anti-inflammatory properties of PG fruit extract and its potential as a photochemopreventive agent (Table 6) [35].

Another in vitro study reported similar effects. After the investigation of the effects of ellagic acid used in concentrations of 25–75 µM in HaCaT cell line, which had been previously exposed to 20 J/cm^2^ UVA radiation, the authors reported a significant dose-dependent inhibition of UVA radiation-induced increases of ROS generation and malondialdehyde production. These results proposed a high efficacy of ellagic acid, present in high concentrations in PG, in protecting human keratinocytes from UVA radiation-induced oxidative stress (Table 6) [36].

UVB radiation (extends from 280 to 320 nm) can initiate DNA damage and mutation of oncogenes and tumor-suppressor genes. Namely, UVB radiation is a complete carcinogen, triggering a photooxidative reaction and increase in the level of ROS [37].

In a study where three-dimensional full-thickness human reconstituted skin (EpiDerm™ FT-200) was used, the effects of different fractions of PG: juice, extract and oil, in UVB-mediated damaged skin cells were analyzed. The results revealed that all three PG fractions in various concentrations inhibited the formation of cyclobutane pyrimidine dimers (CPDs) and 8-dihydro-2′-deoxyguanosine (8-OHdG), which represent important biomarkers of DNA damage induced by UVB. Inhibition of the UVB-mediated protein carbonyl group increase, which represents a protein oxidation marker, was also reported. Furthermore, all three PG fractions inhibited UVB-mediated cell proliferation and increases in tropoelastin levels, inhibited UVB-mediated increases in the protein levels and activity of matrix metalloproteinases (MMPs) and inhibited UVB-mediated phosphorylation of c-jun and expression of c-Fos in the human reconstituted skin model. The results of this study suggested a strong protective effect of PG fractions against UVB radiation- mediated damage and photoaging of human skin (Table 6) [37].

The antioxidant, anti-inflammatory, antiproliferative and DNA-repair effects of the PG fruit extract (acetone/water = 70:30 *v*/*v*) were investigated in an animal skin model (SKH-1 hairless mice), previously exposed to UVB irradiation. After 14 days of UVB exposure (comprising a total of seven sessions) and oral administration of PG fruit extract (0.2%, *w*/*v*), mice were euthanized 24 h subsequent to the final irradiation session, after which the biochemical analysis of harvested skin biopsies was performed. The use of PG fruit extract inhibited multiple UVB-exposure-induced processes, which included hyperplasia, infiltration of leukocytes, protein oxidation, lipid peroxidation, phosphorylation of mitogen-activated protein kinases, activation of NF-κB pathway, cyclooxygenase-2 (COX-2) and inducible nitric oxide synthase (iNOS) protein expression, cyclin D1 and proliferating cell nuclear antigen protein expression and phosphorylation of c-Jun and protein expression of matrix metalloproteinases. These findings imply that oral administration of PG fruit extract also has potentially protective effects on skin exposed to UVB radiation and can be used as a photochemopreventive agent for skin cancer (Table 6) [38].

The effects of PG fruit extract on UVB-mediated oxidative stress and markers of photoaging were also analyzed in an in vitro study using UVB-radiated HaCaT cells. PG fruit extract (10–40 μg/mL) with a polyphenol content of 135,000 p.p.m. gallic acid equivalent and ellagitannins as its predominant constituents was dissolved in 0.1% *v*/*v* dimethyl sulfoxide (DMSO). The PG fruit extract possessed a significant protective effect against UVB-mediated cytotoxicity, which was proven with the increased values of Trolox equivalent antioxidant activity in cells treated with the PG fruit extract. PG fruit extract also prevented the UVB-induced reduction in glutathione levels and tissue inhibitor metalloproteinase 1 levels and the UVB-mediated increase in lipid peroxidase and matrix metalloproteinase protein expression. In addition, PG fruit extract induced the inhibition of UVB-mediated phosphorylation of c-jun and mitogen-activated protein kinases. The outcomes of this investigation indicate that PG fruit extract confers protective effects against UVB-induced oxidative stress and markers associated with photoaging. Thus, the PG fruit extract with photochemoprotective activity could be used in products for shielding against UVB-induced skin damage and the effects of photoaging on the skin (Table 6) [39].

In an in vitro study, which investigated the choice of the primary active ingredient in the cosmeceutical sunscreen products formulation, the PG peel extract was compared with the *Opuntia ficus-indica* peel extract for its photoprotective effects. The authors estimated the sun-protection factor (SPF) with the use of a diffuse reflectance spectrophotometry. The in vitro SPF assay results reported superior effects of the PG peel extract (SPF activity = 44.402 ± 0.438) in comparison to *Opuntia ficus-indica* peel extract (SPF activity = 18.780 ± 0.214). According to the reported SPF values of the PG peel extract, a sunscreen formulation with this primary ingredient can be classified as a product with high sun protection (SPF values ≥ 30). With the high phenol compound content, which act by absorbing UVA and UVB photons and possess antioxidant and antiaging activity, the PG peel extract represents a promising primary active ingredient in cosmeceutical sunscreen formulations (Table 6) [40].

Two extracts, PG and cocoa extract, were analyzed in an in vivo study in SKH-1/CRL mice to determine the effects of extracts on UV radiation-induced skin cancer. The PG peel extract (acetone/water = 70:30 *v*/*v*) contained a total amount of specific phenolic compounds of 12.62% (*w*/*v*). Both extracts used in this study induced marked decreases in the occurrence of skin carcinoma in comparison to the control group that only underwent UV radiation. This was especially pronounced in the group of rats that were treated with PG extract, where very few of the animals exhibited lesions and their skin was classified as normal during histopathological analysis. The PG extract in the dose of 41.9 mg/kg induced a statistically significant difference in skin lesions in comparison to the control group and the group of mice which was treated with cocoa extract. The authors reported lower mutated p53 expression in the treatment groups in comparison to the control group. In addition, a significantly lower proliferating cell nuclear antigen in the group of rats treated with PG extract was reported. These results confirmed the protective effects of the PG extract against UV radiation-induced carcinogenesis (Table 6) [41].

Further, the anticancer activity of the PG fruit extract, which was loaded into solid lipid nanoparticles (SLNs), was analyzed in a cell culture model. The antiproliferative activity of the optimized formulation was assessed by the 3-[4,5-dimethylthiazol-2-yl]-2,5 diphenyl tetrazolium bromide (MTT) assay. The used cell lines included human normal melanocytes (HFB-4) and different cancer cell lines, such as human breast carcinoma (MCF-7), human prostate carcinoma (PC-3) and human hepatocellular carcinoma (HepG2). Researchers reported a 50% inhibitory concentration (IC_50_) of the 49.2 μg/mL PG fruit extract in the optimized formulation on the proliferation of MCF-7 breast cancer cells, IC_50_ of the 107 μg/mL PG fruit extract in the optimized formulation on the proliferation of PC-3 prostate cancer cells and IC50 of the 219 μg/mL PG fruit extract in the optimized formulation on the proliferation of HepG-2 liver cells. Moreover, optimized SLNs were significantly more cytotoxic and effective, in comparison to the free PG fruit extract formulation, as they exhibited a 47-fold reduction in IC50 values for MCF-7 cells, followed by 40-fold for PC-3 cells and 38-fold for HepG-2. The optimized formulation represents a safe delivery system, which enables the enhancement of the efficacy of phytochemicals contained in the PG fruit extract, and consequentially enhances therapeutic effectiveness. These results indicate the possible application of PG extract-loaded SLNs in breast cancer therapy (Table 6) [42].

The anticancer activity, as well as the optimization of the formulation containing the PG extract, were also investigated in a study where the PG fruit extract was incorporated into SLNs, being further added into a transdermal emulgel. The formulation containing a dose of 1.2 mg/mL of PG fruit extract was applied to *Swiss albino* mice with an induced Ehrlich solid carcinoma model. Researchers reported a significant decrease in the tumor volume after a 6-day application of the PG fruit extract-loaded SLNs incorporated into a transdermal emulgel. This delivery system is recognized as a secure and representative method that enhances the bioactivity of phytochemical chemopreventive agents from the PG fruit extract. This improvement is attributed to the higher permeation rate and superior therapeutic effects of the optimized nanoformulation compared to the free PG fruit extract formulation. Transdermal application of the PG fruit extract incorporated in SLNs has great potential in solid breast cancer therapy (Table 6) [43].

### 3.4. Effects on Striae Distensae

Striae distensae are defined as linear, atrophic dermal scars with overlying atrophy of the epidermis. Various types of physiological and pathological conditions can be associated with striae distensae, including obesity, adolescent growth spurts, pregnancy, Cushing’s and Marfan syndromes and long-term use of certain medications, such as topical and systemic corticosteroids. While striae distensae do not typically result in significant medical issues, they represent an aesthetic concern and can cause a psychological burden for many women [44].

The effects of PG seed oil were investigated in an in vivo study, where the authors investigated the potential effects of PG seed oil in combination with *Croton lechleri* resin extract on the improvement of striae distensae. The PG seed oil in the concentration of 4% was incorporated into an emollient oil-in-water cream that was tested in 20 volunteers and its effects were observed comparatively, before and after the treatment. After 6 weeks of topical treatment, investigators reported a significant increase in dermis thickness (14.85%), and positive effects on the hydration (increased by 30.32%) and elasticity (increased by 9.75%) of the *stratum corneum* (Figure 4). Self-assessment reports from volunteers indicated that striae distensae became less defined and less depressed. This study indicates that PG seed oil may prove beneficial in preventing or ameliorating skin changes associated with striae (Table 7) [45].

### 3.5. Effects on Skin Repair Mechanisms

The effects of various fractions of PG (aqueous extracts of PG juice and peel and cold-pressed seed oil) on skin repair mechanisms were analyzed in an in vitro study on skin organ cultures. It was reported that epidermal proliferation was increased by 60% in the presence of 0.5 µL of PG seed oil. The effect that cold-pressed seed oil induced in human skin was a mild thickening of the epidermis in comparison to the control group. Investigating the effects on dermal function, the authors reported the most effective prevention of cell death with the use of the PG peel extract in various concentrations in the range between 0.05 µL/mL and 0.5 µL/mL. In addition, the use of the PG peel extract significantly increased synthesis of type I procollagen. Inhibition of matrix metalloproteinase-1 synthesis by dermal fibroblasts was another reported effect of the PG peel extract. All these reported effects indicate the potential antioxidant, anti-inflammatory and anticarcinogenic activity of the PG peel extract and its effect on the promotion of skin repair. Thus, various parts and extracts of PG could be used for treating different skin damages (e.g., aged skin and photoaged skin bruises or chronic bruises), which not only pose a cosmetic concern but also contribute to considerable morbidity (Table 8) [46].

In an in vivo animal study, the wound-healing potential of PG peel extract was analyzed. An ethanol extract of PG peel was integrated into an ointment formulation in a concentration of 5% (*w*/*w*), and applied for the treatment of excision wounds on the dorsal interscapular area of guinea pigs. Researchers reported faster wound contraction rate in the group treated with the PG peel extract ointment with complete healing and no scarring after 20 days of treatment, due to the enhanced epithelization process, antioxidant activity and increase in hydroxyproline, total DNA and protein content on the injured site. These results indicate the potential use of PG in wound healing (Table 8) [47].

In another study that analyzed the wound-healing activities of PG peel extract, the authors used an animal model of *Wistar* rats with induced second-degree burns. Animals were treated with 100 mg/mL ethanol PG peel extract (formulated as a crude extract after the evaporation of ethanol and reconstituted with water before use) twice daily for 21 days. Researchers reported faster wound healing in the group of rats treated with the PG peel extract with a 97% contraction of wounds in comparison to the group of rats treated with the silver–sulfadiazine ointment (79% contraction of wounds). These findings led to the conclusion that PG peel extract, as a safe and effective treatment option, can be used for treating burns as an alternative to silver-based preparations that can cause various side effects (Table 8) [48].

A case study of a 76-year-old female patient with complaints about a non-healing ulcer on the left leg, accompanied by pain and swelling, also reported the effects of PG peel extract on wound healing. The reported ulcer was characterized by shiny granulation tissue and did not respond to any of the conventional therapy options. After 6 weeks of application of a hydrophilic cream containing zinc oxide in addition to ethanolic PG peel extract (2% *w*/*w*) once daily, the ulcer had reduced to one-quarter of its original size, and after 90 days of application, complete healing was reported (Figure 5). This case study indicated that the PG peel extract has the potential to be incorporated in various preparations that can expedite the wound-healing process (Table 8) [49].

### 3.6. Antiaging Effects

In a double-blind, placebo-controlled pilot clinical trial, the effects of orally administered PG extract on skin wrinkles, biophysical characteristics and the gut–skin axis were investigated. The trial included 18 participants of both genders, aged 25–55 years. High-resolution facial photographs were utilized for assessing wrinkles, with the average severity calculated based on depth and width measurements. Additionally, non-invasive instruments were used to measure the biophysical properties of the skin, including epidermal water loss, facial sebum production, facial erythema index and melanin index. Facial skin swabs were collected for DNA extraction and quantification, followed by bioinformatics analysis to evaluate changes in the skin microbiome. The results revealed a statistically significant decrease in facial wrinkle severity (by 6.2 ± 1.6% in the PG extract group) and a reduction in sebum excretion rate on the forehead (by 21.9 ± 14.4% in the PG extract group). However, there were no statistically significant changes observed in facial melanin or erythema indices, and no shift in the skin microbiome was detected, as indicated by the absence of an increase in the relative abundance of *Staphylococcus epidermidis* and *Bacillus genus species*, after a 4-week period of oral administration of the PG extract. Moreover, no adverse effects were observed during the clinical trial. The outcomes of this investigation confirmed the positive effects of the PG extract on the skin barrier function and transepidermal water loss, which resulted in enhancing the skin barrier function and improving wrinkle appearance and severity. These reported results correlate with the potent antioxidant and anti-inflammatory features of the phytochemicals present in the PG extract (Table 9) [50].

### 3.7. Effects on Skin Erythema and Pigmentation

In a single-blinded, placebo-controlled clinical trial, the effects of the PG peel extract (incorporated in an oil-in-water microemulsion) on skin changes regarding erythema and melanin pigmentation of volunteers were assessed. After the application of the microemulsion as a night routine for a period of 12 weeks, the authors reported a statistically significant continuous decrease in skin melanin content in subjects, probably due to the antioxidant activity of the extract. Moreover, a statistically significant reduction in skin erythema was reported by the group of subjects using the microemulsion with the incorporated PG peel extract, probably due to the antioxidant and anti-inflammatory effects of the extract. These results suggested that PG peel extract can be incorporated into a stable microemulsion possessing significant skin compatibility characteristics, which can be further applied in the management of skin pigmentation disorders (Table 10) [51].

Furthermore, a double-blind, randomized, placebo-controlled, split-faced study analyzed the skin-lightening effects of PG peel extract in 30 volunteers. The phenolic-rich PG peel extract was prepared in the form of a serum and mask and applied by the volunteers for 28 consecutive days. The preliminary investigation of the skin irritation by the use of the closed-patch method indicated the complete safety of the products. Over the study period, volunteers had no adverse effects, which confirmed the safety of the PG peel extract serum and mask. Facial skin lightening and a significant reduction in the melanin content were reported after the use of the serum and mask containing 0.2% of the PG peel extract due to its antimelanogenesis effect. The positive effects on hyperpigmentation were shown after one week and the maximum effect of both formulations was reported after 28 days. Moreover, the authors reported the serum formulation as the better dosage-form, which contributes to the PG peel extract effects. These results indicated that the PG peel extract possesses great potential for incorporation in various cosmetic formulations, which are safe and effective for use in different skin pigmentation disorders (Table 10) [52].

### 3.8. Effects on Psoriasis

Psoriasis represents a chronic immune-mediated disease. A key role in the pathogenesis of this condition is played by the interaction of T cells and keratinocytes, and the interleukin (IL)-23/IL-17 axis is deemed to be vital [53]. Approximately 90% of psoriasis cases are cases of chronic plaque-type psoriasis (psoriasis vulgaris). This condition is classified by sharply defined, erythematous, itchy patches adorned with silvery scales, which can extend over significant portions of the skin on various body regions [54].

In a study that analyzed the effects of punicalagin, the main chemical compound isolated from PG peel, on psoriasis, the authors used the imiquimod-induced psoriasis model in mice, as well as TNF-α- and IL17A-stimulated HaCaT cells. The animals underwent treatment with a topical dose of 25 mg/kg of punicalagin (incorporated into a carbomer gel also containing penetration enhancers, i.e., azone and ethanol) twice a day for 7 consecutive days. Improved psoriatic-related phenotypes with the use of punicalagin gel were obtained, which effectively relieved the severity of imiquimod-induced psoriatic-like symptoms. In vitro analysis of punicalagin anti-inflammatory activity in the inflammatory psoriatic model in HaCaT cells was further investigated. Dose-dependent effects of punicalagin on the reduction of the upregulated IL-1β gene and pro-/mature protein expression through suppressing the activation of NF-κB and the expression of caspase-1 were seen. Moreover, the punicalagin-induced reduction of IL-1β transcription and secretion were observed in vivo, which indicates the possibility of punicalagin being used for the alleviation of psoriasis symptoms through the IL-1β targeting mechanism that represents a promising strategy for treating psoriasis (Table 11) [55].

### 3.9. Effects on Hair Color Protection

Antioxidant activity of the PG peel extract could be used in cosmetics, such as for hair color protection, because of its potential to reduce color change due to the prevention of the oxidation process. A study was performed that analyzed the antioxidative effects of the PG peel extract incorporated into liposomes (Table 1). The authors reported an over 90% inhibition of DPPH radicals with the use of the PG peel extract formulation, while the encapsulation of the extract into a liposomal formulation further increased the antioxidant activity. Furthermore, the PG peel extract exhibited the highest Trolox equivalent antioxidant capacity (TEAC) value. However, a small decrease in TEAC value was observed for the liposomes, likely attributed to the partial degradation of bioactive ingredients during the manufacturing. After the analysis of dyed human hair bundles, greater hair color protection with the use of liposomal formulation of the PG peel extract was reported. These results confirmed the high antioxidant activity of the PG peel extract which can be further enhanced by incorporating the extract into liposomes. Moreover, the PG peel extract in the liposomal formulation can enhance hair color protection by the formation of a lipid-based film layer on hair fibers. This formulation could be used in different types of cosmetic products, such as shampoos, hair conditioners and hair serums [22].

The potential use of PG in dermatology and cosmetics is based on the reduction of oxidative damage, which could be used in therapies for dermatological disorders as well as for the prevention of premature ageing. Other important effects of phytochemical compounds with antioxidant activity are their photoprotective and anti-inflammatory activities that are useful for the treatment of sun-stressed and sensitive skin. With the use of antioxidants in cosmetic products, the reduction in oxidative deterioration and oxidation of oily components of the formulation is also ensured. Thus, antioxidants have positive effects on skin health and on formulation stability [23].

## 4. Discussion

Different parts of the PG plant contain various active chemical compounds. After being taken orally, these phytochemicals can induce different pharmacological effects aimed at addressing various conditions. Clinical trials have substantiated the therapeutic potential of PG and its phytochemicals in addressing various diseases, encompassing diabetes, other endocrine disorders, cardiovascular disorders, and cancer. The demonstrated anti-inflammatory effects; protective mechanisms against hyperlipidemia and hypertension; and inhibition of cancer proliferation and diabetes underscore its promising medicinal attributes. More clinical trials are necessary to precisely determine the underlying molecular mechanism of active PG compounds in these conditions [56]. Besides numerous positive effects after oral administration, different parts of PG are used also for various dermatological conditions, including diseased, but also healthy skin. Research results suggest the dermatological use of different parts of the plant and report successful treatment outcomes in various skin conditions. The present review provides an insight into the available studies that analyzed different mechanisms of action of PG in several skin conditions, as well as the effects of various plant materials obtained from PG.

Antioxidant activity, which is reported in several studies which analyzed the effects of PG seed, fruit and peel extracts, indicates the high antioxidant effects of PG that can be very useful in cosmetic products for skincare, as well as for hair color preservation. According to previous studies, forthcoming dermatological and cosmetic formulations containing PG extract or PG juice incorporated into gold nanoparticles are anticipated to augment the established antioxidant efficacy [57].

The complex mechanisms of the anti-inflammatory effect of PG fruit extract include its cytotoxic activity on inflamed cells, the inhibition of production of several proinflammatory cytokines and the suppression of nitric oxide production. All these properties can be useful for treating and alleviating the symptoms of inflammatory skin conditions.

Antimicrobial activity against several types of fungus and bacteria indicated the potential of PG peel, flower and seed extract and their active compounds for the treatment of acne vulgaris, oral candidiasis, chronic dandruff and several other types of bacterial or fungal infections. Antimicrobial activity was confirmed in a randomized, double-blinded clinical study where mouthwash containing PG flower extract was reported to be safe and effective against bacteria that caused gingivitis [58].

A vast number of in vitro studies or in vivo animal studies investigated chemoprotective activity against UVA and UVB radiation-induced skin damage. These effects were proven through various mechanisms, all suggesting a strong potential of the PG active compounds. Thus, PG extracts should be incorporated into formulations of different cosmetic products that are used for UV protection and cancer adjunctive therapy. Various mechanisms of photoprotection in natural molecules derived from different plant extracts are suitable for incorporation in sunscreen formulations. PG extracts and their constituents have shown large potential as primary active ingredients in these formulations, as their photoprotective activity is comparable with the photoprotective activities of other natural products, such as rutin, quercetin, caffeine, rosmarinic acid, etc. [59]. Due to the increased interest in natural product incorporation in the next generation of sunscreen products, more in vivo studies are needed to assess their efficacy and safety. Although in vivo investigations proved the photoprotective effects, chemoprotective effects and/or safety of different natural compounds or extracts, such as resveratrol, rutin, ferulic acid, caffeine, Merostachys pluriflora bamboo culms extract, Lippia sericea extract and Bauhinia microstachya extracts, as well as industrial process waste material such as Cabernet sauvignon pomace extract and Agave sisalana Perrine sisal [60,61], in vivo investigations regarding the effects of PG extracts in sunscreen cosmetic products have not been recorded.

Striae distensae represents a significant aesthetic skin condition in which PG seed oil was reported to be effective. More investigations into different formulations containing PG seed oil should be conducted to define a safe and effective cosmetic product that would be useful in both the prevention and improvement of skin changes associated with striae.

The PG juice, peel and seed extracts exerted positive effects on skin repair mechanisms in various conditions, such as photodamaged skin, bruises and second-degree burns. Formulations with adequate concentrations of different PG fractions can be used in treating these skin changes and the precise mechanisms involved in the skin repair mechanism should be evaluated.

The antiaging effect of PG extract was proven in a clinical trial where significant skin barrier improvement and decreased wrinkle appearance and severity were obtained, further showing the strong potential of PG active compounds in antiaging cosmetic products.

Human subjects treated with formulations containing PG peel extract experienced a noteworthy decrease in skin erythema. Incorporating this extract into their nightly skincare routine presents a promising, safe and effective option for addressing various types of skin pigmentation.

Psoriasis, an autoimmune skin condition, requires multidisciplinary treatment involving various medications and adjunctive therapies. In vivo studies on animals demonstrated the efficacy of PG peel extract in alleviating psoriatic-like symptoms. This underscores the potential of incorporating PG peel extract into formulations to relieve skin-related symptoms, necessitating further analysis to identify the most effective formulation for this purpose.

The irritation potential, as well as the other possible side effects of ingredients used in cosmetic products, should be kept in mind. None of the abovementioned studies reported any side effects of PG extracts or oil. This information was confirmed in a safety assessment of the PG-derived ingredients which are used in cosmetic products. Furthermore, according to the FDA, essential oils, oleoresins (solvent-free) and natural extracts (including distillates) derived from PG are generally recognized as safe (GRAS). There are no available toxicity data that indicate side effects of any of these cosmetic ingredients derived from PG [62]. Investigations related to the claimed health-beneficial properties of PG are on the rise in recent decades. Conversely, its safety profile and potential adverse effects have not been systematically investigated, primarily due to the classification of pomegranate and its derivatives as food products. However, according to the 2019 Cosmetic Ingredient Review Expert Panel, the available dermal irritation and sensitization data on the fruit extract are deemed adequate, suggesting its safe use. Also, according to the World Health Organization monograph for *Pericarpium Granatum*, no genotoxic effect was shown. The preparation and administration of PG end-products are influenced by numerous factors. Their documentation should be meticulously prepared and integrated into regulatory guidelines, ensuring strict control before PG products are permitted for sale on the market [62].

## 5. Conclusions

This review summarized the current limited knowledge about the effects of various *Punica granatum* extracts on healthy and diseased skin. While the research on the effects of *Punica granatum* on skin health remains limited, the available evidence, being represented here, suggests promising avenues for its potential application in managing various skin conditions. Despite the gaps in knowledge, the diverse range of reported positive effects of *Punica granatum* on overall health and few, but promising, data of the effects of *Punica granatum* on skin health underscores its potential significance in the care of healthy and diseased skin. Further exploration of its mechanisms of action and targeted studies are warranted to fully understand and harness its therapeutic benefits for skincare. By bridging these gaps in research, we can unlock the valuable potential of *Punica granatum* as a natural remedy for promoting skin health and addressing dermatological concerns. Additionally, long-term studies examining the prolonged effects of *Punica granatum* on skin health could provide valuable insights into its sustainability and long-term benefits. Moreover, exploring interactions of *Punica granatum* with other skincare ingredients and its compatibility with different skin types would contribute to a more comprehensive understanding of its role in dermatological care. Future research endeavors should focus on elucidating the optimal formulations and concentrations of *Punica granatum* for specific skin conditions to maximize its efficacy and safety in clinical practice.

## 6. Future Directions

The current review provides a comprehensive overview of the botany, phytochemistry and dermatological effects of *Punica granatum*, with a specific emphasis on its therapeutic mechanisms in various skin conditions. The abundant presence of polyphenolic compounds, particularly in PG peel, imparts significant antioxidant properties, making it a promising candidate for skin health promotion. While in vitro and in vivo studies have shown promising results regarding the beneficial effects of active PG compounds on skin health, there is a need for more extensive in vivo investigations involving human volunteers. These studies would help validate the findings obtained from laboratory experiments and identify optimal concentrations and formulations for specific skin conditions.

Further exploration of the mechanisms underlying the dermatological effects of *Punica granatum* is essential for fully harnessing its therapeutic potential. Targeted studies elucidating its mode of action can provide valuable insights into its efficacy and safety profile, facilitating its integration into clinical skincare practices. Thus, the mechanisms of the effects of *Punica granatum* should be investigated more intensively in future.

Long-term studies assessing the prolonged effects of *Punica granatum* on skin health are warranted to ascertain its sustainability and enduring benefits. These investigations would shed light on its viability as a long-term skincare remedy and its potential role in preventive dermatology.

Exploring interactions between *Punica granatum* and other skincare ingredients in order to elucidate their synergistic effects or potential incompatibilities is necessary. In addition, its compatibility with various skin types is crucial for a comprehensive understanding of its role in dermatological care. Such investigations can inform formulation strategies tailored to diverse skin needs and preferences.

Future research endeavors should focus on determining the optimal type of formulations and concentrations of *Punica granatum* for specific skin conditions to maximize efficacy and safety in clinical settings. This optimization process would facilitate its integration into evidence-based dermatological therapies.

In summary, by addressing these future directions, we can unlock the full potential of *Punica granatum* as a natural remedy for promoting skin health and addressing dermatological concerns, thereby enhancing overall well-being and quality of life.

## Figures and Tables

**Figure 1 pharmaceutics-16-00458-f001:**
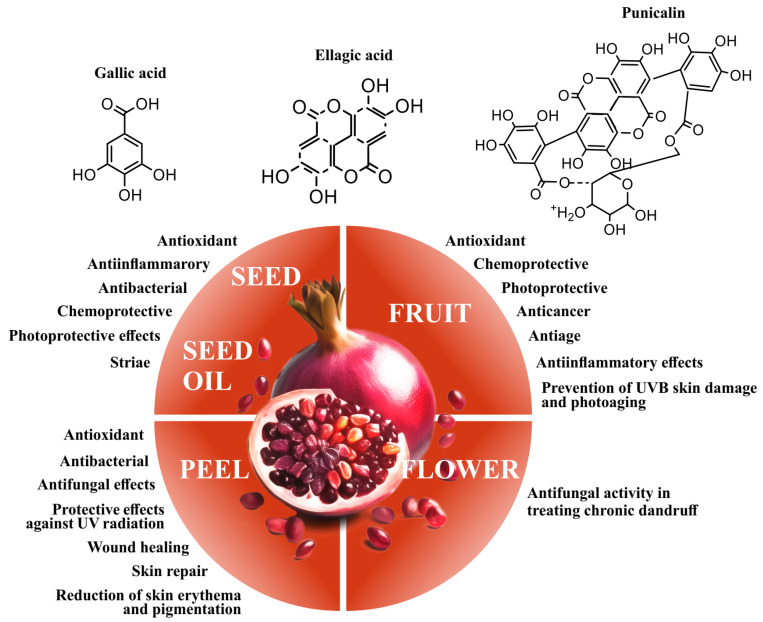
Skin protection induced by *Punica granatum*.

**Figure 2 pharmaceutics-16-00458-f002:**
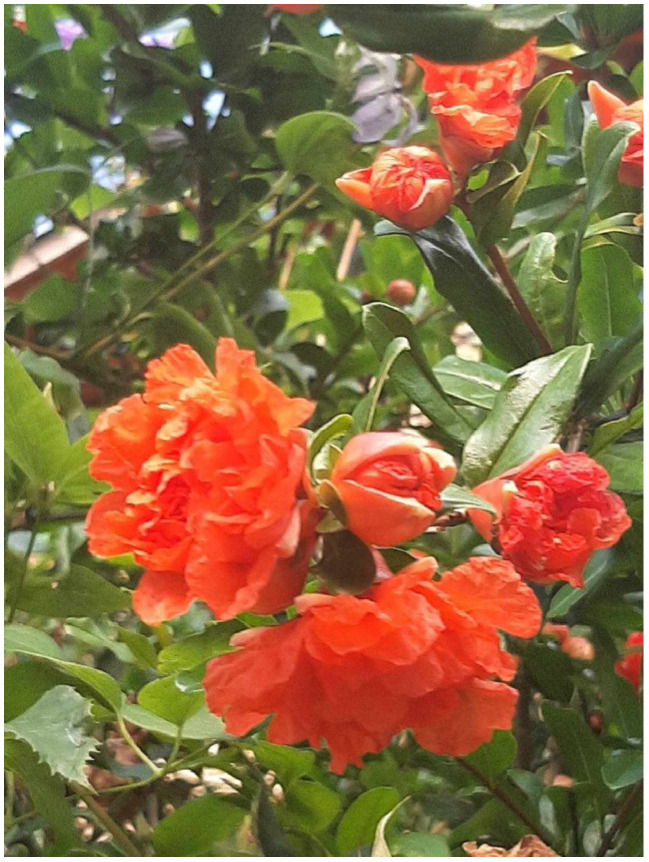
The *Punica granatum* flower.

**Figure 3 pharmaceutics-16-00458-f003:**
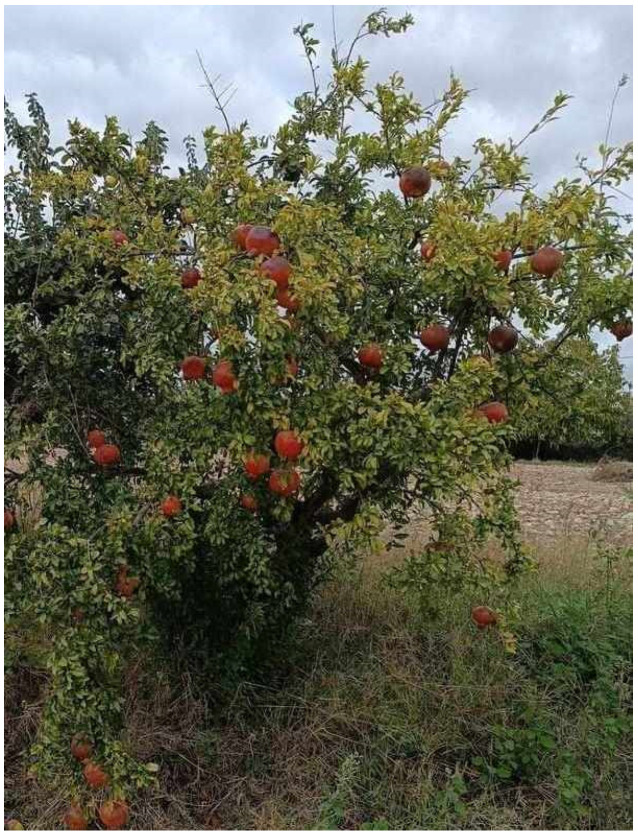
The *Punica granatum* tree.

**Figure 4 pharmaceutics-16-00458-f004:**
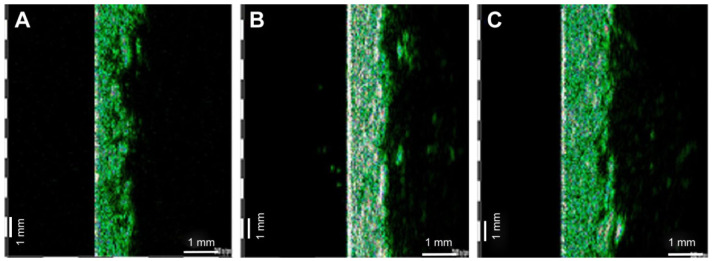
Ultrasonographic imaging of the skin before (**A**), after 3 weeks (**B**) and after 6 weeks (**C**) of cream application, revealing the new echogenic areas instead of striae distensae, as well as the improvement in the acoustic homogeneity of the dermis [45]. *Drug Design, Development and Therapy 2017:11 521–531*—originally published by and used with permission from Dove Medical Press Ltd (Macclesfield, UK).

**Figure 5 pharmaceutics-16-00458-f005:**
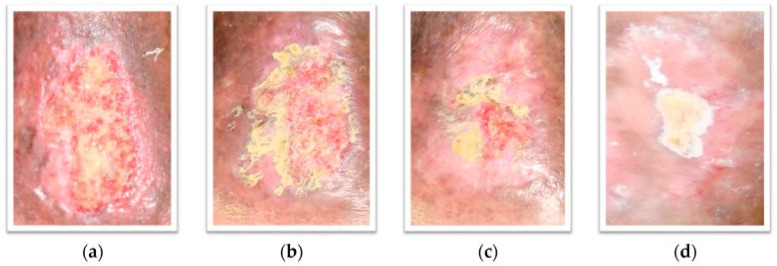
Photographs of the patient’s wound before (**a**), after 1 week (**b**), after 6 weeks (**c**) and after 10 weeks of treatment (**d**). Originally published by and used with permission from MDPI [49].

**Table 1 pharmaceutics-16-00458-t001:** Active compounds isolated from the PG peel extract and PG juice.

Compounds	Concentration in the PG Peel Extract	Concentration in the PG Juice	Study
Total phenolics	320.16 ± 2.39 mg GAE/g DW	146 ± 2.01 mg GAE/100 mL juice	Study 1 Živković et al. [13]
Total tannins	17.37 ± 0.64%	5.21 ± 0.22%	
Punicalagin	54.23 ± 1.65 mg/g DW	43.23 ± 1.45 mg/g DW	
Punicalin	197.13 ± 3.48 mg/g DW	86.93 ± 2.45 mg/g DW	
Gallic acid	6.83 ± 0.53 mg/g DW	6.69 ± 0.33 mg/g DW	
Ellagic acid	25.42 ± 0.80 mg/g DW	6.49 ± 0.71 mg/g DW	
Punicalagin A	13.09 ± 1.33 μg/mL	56 ± 13.48 μg/mL	Study 2 Shahkoomahally et al. [14]
Punicalagin B	13.33 ± 1.36 μg/mL	83.09 ± 36.77 μg/mL	
Ellagic acid	399.37 ± 44.48 μg/mL	6.42 ± 1.32 μg/mL	
Gallic acid	22.54 ± 2.96 μg/mL	1.05 ± 0.33 μg/mL	

GAE—gallic acid equivalent; DW—dry weight.

**Table 2 pharmaceutics-16-00458-t002:** Antioxidant effects of *Punica granatum*.

*Punica granatum* Formulation	Effect	Model	Material	Dosage/ Concentration	Mechanism	Study
Cold pressed seed oil	Antioxidant and anti-inflammatory activity	Modification of Hammerschmidt and Pratt method	Cyclooxygenase from sheep vesicula seminalis, soybean lipoxygenase	5.3% of the dry weight consisted of isolated punicic acid (65.3%), accompanied by palmitic acid (4.8%), stearic acid (2.3%), oleic acid (6.3%), and linoleic acid (6.6%)	Inhibition of enzymes in the eicosanoid pathway, cyclooxygenase and eicosanoid pathway enzyme, lipoxygenase	Schubert et al. [19]
Fruit extract	Antioxidant activity	Measuring of ROS and levels of caspases	HaCaT cells	6.25–100 μg/mL	Decrease the level of hydrogen peroxide-induced apoptosis and downregulation of caspase-3 and caspase-7	Liu et al. [20]
Seed extract, peel extract	Antioxidant activity	TBA method, hydroxyl radical scavenging activity and LDL oxidation	β-Carotene-Linoleate and DPPH Model System, TBA Assay	1.04%, 9.38%, 7.53% peel extract 2.32%, 8.62%, 7.53% seed extract	Synergism of different antioxidant mechanisms	Singh et al. [21]
Peel extract incorporated in liposomal formulation	Antioxidant activity	Measuring of DPPH scavenging activity and TEAC values and ex vivo effects	DPPH model system, TEAC assays, dyed human hair bundles	0.75 g	Synergism of different antioxidant mechanisms increased by formation of a lipid-based film layer on the hair fibers by liposome particles	Tanriverdi et al. [22]

**Table 3 pharmaceutics-16-00458-t003:** Anti-inflammatory activity of *Punica granatum*.

*Punica granatum* Formulation	Effect	Model	Material	Dosage/Concentration	Mechanism	Study
Ethyl acetate isolated ellagic acid, gallic acid and punicalagins A and B	Anti-inflammatory activity	LPS-stimulated cell culture	RAW264.7 cells	50–200 μg/mL	Inhibition of nitric oxide (NO) production that is dependent on the dose and suppression of PGE-2 and IL-6 production	BenSaad et al. [15]

**Table 4 pharmaceutics-16-00458-t004:** Antimicrobial activity of *Punica granatum*.

*Punica granatum* Formulation	Effect	Model	Material	Dosage/Concentration	Mechanism	Study
Peel extract	Antifungal activity	Induced oral candidiasis	*Wistar* rats	125 µg/mL, 250 µg/mL, 500 µg/mL	Inhibition of *C. albicans* growth	Bassiri- Jahrom et al. [24]
Peel and seed extract	Antibacterial activity	Measuring of MIC and MBC, time-dependent growth inhibition assay	MRSA strains	1.25 mmol/L	Delay of initial growth of MRSA	Hussein et al. [25]
Peel extract	Antibacterial and antifungal activity	Diffusion disc method, broth microdilution method	Various types of pathogens	2.7 mg/mL	Inhibition of growth of different Gram-negative and Gram-positive bacteria and fungal pathogens	Kupnik et al. [26]
Methanolic flower extract	Antifungal activity in treating chronic dandruff	Clinical trial	Human subjects	28.1% (*w*/*w*)	Antioxidant activity, anti-itching and anti-inflammatory (inhibition of COX, LOX and PLA2)	Sahraie-Rad et al. [27]

**Table 5 pharmaceutics-16-00458-t005:** Effects of *Punica granatum* on acne vulgaris.

*Punica granatum* Formulation	Effect	Model	Material	Dosage/Concentration	Mechanism	Study
Acetone extract	Antibacterial activity	Testosterone-induced accumulation of epithelial cells and keratin, anti-lipase activity for simulation of sebum accumulation	*Wistar* rats, HaCaT cells, heat-killed *P. acnes*	0.1–10 mg/site, 0.25–1 mg, 2.5–10 mg	Shrinkage and damage of *P. acnes* and *S. aureus* and inhibition of lipase activity	Lee et al. [30]

**Table 7 pharmaceutics-16-00458-t007:** Effects of *Punica granatum* on striae distensae.

*Punica granatum* Formulation	Effect	Model	Material	Dosage/Concentration	Mechanism	Study
Seed oil in emollient oil-in-water cream	Improvement in skin changes associated with striae	Clinical trial	Human subjects with detected striae distensae	4%	Increase in dermis thickness, hydration and elasticity of the *stratum corneum*	Bogdan et al. [45]

**Table 8 pharmaceutics-16-00458-t008:** Effects of *Punica granatum* on skin repair mechanisms.

*Punica granatum* Formulation	Effect	Model	Material	Dosage/Concentration	Mechanism	Study
Peel extract, cold-pressed seed oil	Promotion of skin repair	In vitro study of skin organ culture	Human skin cells	0.05 µL/mL and 0.5 µL/mL; 0.5 µL/mL	Inhibition of matrix metalloproteinase-1 synthesis by dermal fibroblasts, antioxidant, anti-inflammatory and anticarcinogenic activity	Aslam et al. [46]
Methanolic peel extract incorporated in ointment preparation	Wound-healing properties	Excision wound on the dorsal interscapular area	Guinea pigs	5% (*w*/*w*)	Enhancing epithelization process, antioxidant activity increase in hydroxyproline, total DNA and protein content on the injured site	Hayouni et al. [47]
Crude peel extract reconstituted with water before use	Wound-healing properties	Second-degree burn model	*Wistar* rats	100 mg/mL	Reduction of inflammatory cell infiltration and antibacterial activity	Ma et al. [48]
Ethanolic peel extract incorporated in hydrophilic cream	Wound-healing properties	Case presentation of a 76-year-old female patient	Non-healing ulcer characterized by shiny granulation tissue	2% (*w*/*w*)	Complete healing of the ulcer due to biological activities of various active compounds of peel extract	Fleck et al. [49]

**Table 9 pharmaceutics-16-00458-t009:** Antiaging effects of *Punica granatum*.

*Punica granatum* Formulation	Effect	Model	Material	Dosage/Concentration	Mechanism	Study
Oral fruit extract, standardized to punicalagin	Antiaging, antioxidative and anti-inflammatory effects	Clinical trial	Human subjects	75 mg	Decrease of transepidermal water loss, enhancing skin barrier function and improving wrinkle appearance and severity	Chakkalakal et al. [50]

**Table 10 pharmaceutics-16-00458-t010:** Effects of *Punica granatum* on skin erythema and pigmentation.

*Punica granatum* Formulation	Effect	Model	Material	Dosage/Concentration	Mechanism	Study
Peel extract incorporated into oil-in-water microemulsion	Reduction of skin erythema and pigmentation	Clinical trial	Human subjects	No information	Antioxidant activity of ellagic acid, antioxidant and anti-inflammatory activity of constituents of peel extract	Parveen et al. [51]
Peel extract incorporated into facial serum and mask	Reduction of skin pigmentation	Clinical trial	Human subjects	0.2%	Antimelanogenesis activity of the phenolic compounds	Kanlayavattanakul et al. [52]

**Table 11 pharmaceutics-16-00458-t011:** Effects of *Punica granatum* on psoriasis.

*Punica granatum* Formulation	Effect	Model	Material	Dosage/Concentration	Mechanism	Study
Punicalagin	Possible alleviation of psoriasis symptoms	Imiquimod-induced psoriasis model, TNF-α- and IL17A-stimulated cell culture	BALB/c mice, HaCaT cells	25 mg/kg	Downregulation of IL-1β gene and protein expression via NF-κB and caspase-1 inhibition in vitro, and reduced IL-1β transcription and secretion in vivo.	Tang et al. [55]

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
