# Peer review of "Punica granatum L. (Pomegranate) Extracts and Their Effects on Healthy and Diseased Skin"

_pharmaceutics, 2024, doi:10.3390/pharmaceutics16040458_

Round 1

Reviewer 1 Report

Comments and Suggestions for Authors

The authors realized an interesting article showing a summary of the phytochemistry, botany, and dermatological effects of Punica granatum (PG), emphasizing on therapeutic mechanisms in different skin conditions but some modification should be made in order to increase the quality of the article:

1. Regarding the Figure 1 it should be improved the quality, to enlarge the written part of the structures in order to increase the readability.

2. The botany description it should be more detailed.

3. The active compounds should be introduced in a table in order to be easier to read.

4. Table 1 should be reorganized completely because it is not suitable with the author’s guidelines for the journal Pharmaceutics. Also it is not acceptable that one entire page should have only 4 rows of table. I propose that the entire Table 1 should be split in other mini-tables according with the action: 1 table at the antioxidant activity, 1 table at the antimicrobial activity…etc..

5. When writing species please write them in italics. Ex: P. acnes and S. aureus

6. Add more figures describing the mode of action in each skin disorder and emphasizing on the beneficial effects.

7. The entire manuscript should be reformulated in order to decrease the percent match below to 20 %. Now it is at 36%.

Author Response

Dear Reviewer,

Thank you for taking the time to read and comment on our manuscript. The authors have revised the manuscript according to your comments with the revised parts marked in red color.

Following your advice, we have corrected the weak points within the article and we believe that this has markedly improved the quality of the manuscript.  

Reviewer 2 Report

Comments and Suggestions for Authors

The content of the manuscript is interesting, the manuscript offers an overview of the possible application of different parts of the fruit, but as it is stated in the introductional section, very few applications have been carried out.

The main reason is the lack of safety information. It is noteworthy to highlight that medicinal products are subject to stringent quality testing to ensure safety and efficacy in use, whereas there are no comparable regulatory standards and specific labeling requirements for dietary supplements mentioned in the present work. When using herbal products, compliance with established standards in health research is essential. The lack of that information is the main obstacle for publishing the work in the journal like Pharmaceutics.

The review on the botany, phytochemistry and dermatological effects for different skin issues is well presented, but anyhow, several contributions exist on the same topic.

Next; the references are not up to date; several recent references are missing; such as https://doi.org/10.3390/plants11070928.

In my opinion, this review is not suitable for publication in the journal of such ranking.

Comments on the Quality of English Language

The content of the manuscript is interesting, the manuscript offers an overview of the possible application of different parts of the fruit, but as it is stated in the introductional section, very few applications have been carried out.

The main reason is the lack of safety information. It is noteworthy to highlight that medicinal products are subject to stringent quality testing to ensure safety and efficacy in use, whereas there are no comparable regulatory standards and specific labeling requirements for dietary supplements mentioned in the present work. When using herbal products, compliance with established standards in health research is essential. The lack of that information is the main obstacle for publishing the work in the journal like Pharmaceutics.

The review on the botany, phytochemistry and dermatological effects for different skin issues is well presented, but anyhow, several contributions exist on the same topic.

Next; the references are not up to date; several recent references are missing; such as https://doi.org/10.3390/plants11070928.

In my opinion, this review is not suitable for publication in the journal of such ranking.

Author Response

(The authors gave the same response as above.)

Reviewer 3 Report

Comments and Suggestions for Authors

Dear Authors,

I write you in regard to your review manuscript entitled "Punica granatum L. (pomegranate) plant extracts in the treatment of healthy and diseased skin". The motivation for your review is of interest to the pharmaceutical sciences, including the pharmaceutical and cosmetic industries. One major concern was the necessity to turn clear what would be a cosmetic or pharmaceutical use of the pomegranate derivatives. For instance, the word treatment should not be used for a healthy skin approach to the pomegranate extracts, since it could generate a misunderstanding between cosmetics and medicines. Perhaps, defining them could be an interesting idea to avoid misinterpretation. Figure 1 could be revised according to what we previously suggested. Thus, the title must be revised. Still considering the title, it appeared not to be in alignment with the objectives in lines 51-53. Please, add figures of the plant to illustrate sections 1.1 and 1.2. 

If possible, try separating the cosmetic use from the medical use of the pomegranate derivatives from section 2. 

Please, try revising Table 1. It was unnecessarily extensive. Perhaps, to add value to the review, separate extracts from technologies. 

Section 3.1 seemed confusing. No test to prove the anti-acne was reported. Please, present clinical trials or correlated assays. 

Sections 3.4 and 3.5 reported the same issue.

The part of the review regarding photoprotection must be updated with recent papers that, even though this plant was not used as mentioned, suggest its application from the literature that uses natural compounds (rutin, caffeine, rosmarinic acid etc.) to increase the efficacy of sunscreens proved in vivo in humans.

The conclusion must answer the objectives. Please, revise this section.

Author Response

(The authors gave the same response as above.)

Round 2

Reviewer 1 Report

Comments and Suggestions for Authors

Reggarding the figure 4, the name of the figure should be after the figure not before and it should be explained what is the green layer?

Figure 5 the same, it should be written after the figure the legend. 

Author Response

Thank you for taking the time to read and comment on our manuscript. The authors have revised the manuscript according to your comments with the revised parts marked in blue color.

Following your advice, we have corrected the weak points within the article and we believe that this has markedly improved the quality of the manuscript. 

Reviewer 2 Report

Comments and Suggestions for Authors

In my opinion, the manuscript does not meet the requirements to be published in Pharmaceutics. Table 1 should be edited according to the author’s guidelines for the journal Pharmaceutics.

I would ask for the supplementary file as the marked version of the manuscript to identify the modifications after the first revision.

The complete structure of the manuscript does not fit to the form proposed in the guidelines.

In my opinion, the scope of the manuscript fits well to the one of the journal, but the extent of the review is not sufficient. Besides, many information is basic and well known .

Author Response

(The authors gave the same response as above.)

Reviewer 3 Report

Comments and Suggestions for Authors

Dear Authors,

Thank you for improving your review manuscript. Please, try to consult and incorporate references for the photoptoection issue involving in vivo assays. The reference 59 did not fit into this subject, although, it is an interesting reference. Please, consider: https://doi.org/10.1111/ics.12890; and https://doi.org/10.1111/jocd.13609.

Author Response

(The authors gave the same response as above.)

Round 3

Reviewer 2 Report

Comments and Suggestions for Authors

The authors have addressed the reviewers comments so that the quality of the manuscript has been improved considerably.